# Regulation of Death Receptor Signaling by S-Palmitoylation and Detergent-Resistant Membrane Micro Domains—Greasing the Gears of Extrinsic Cell Death Induction, Survival, and Inflammation

**DOI:** 10.3390/cancers13112513

**Published:** 2021-05-21

**Authors:** Jürgen Fritsch, Vinzenz Särchen, Wulf Schneider-Brachert

**Affiliations:** 1Department of Infection Prevention and Infectious Diseases, University Hospital Regensburg, Franz-Josef-Strauß-Allee 11, 93053 Regensburg, Germany; Wulf.Schneider@ukr.de; 2Institute for Experimental Cancer Research in Pediatrics, Goethe-University, 60528 Frankfurt, Germany; v.saerchen@kinderkrebsstiftung-frankfurt.de

**Keywords:** palmitoylation, membrane compartments, TNF-R1, CD95, TRAIL-R1, death receptor

## Abstract

**Simple Summary:**

Death receptor activation can induce various signaling cascades ranging from cell survival to different forms of regulated cell death. The diversification of these biological outcomes frequently depends on the subcellular localization of the receptors. Activation by their ligands at the plasma membrane can change their plasma membrane localization to form distinct receptor–ligand bound signaling complexes. Receptors can also be internalized to signal from endosomes or the nuclear compartment. Those signaling complexes can be further remodeled en route and are partially released to signal from the cytoplasm. Reversible post-translational modification via S-Palmitoylation, a form of lipidation, emanated as a major regulator of death receptor signaling over the past years. We highlight what is known about S-Palmitoylation in different receptor systems, how it affects localization of the receptor complexes in specialized membrane micro domains, and the functional consequences and therapeutical potential of altered S-palmitoylation in the respective signaling cascades.

**Abstract:**

Death-receptor-mediated signaling results in either cell death or survival. Such opposite signaling cascades emanate from receptor-associated signaling complexes, which are often formed in different subcellular locations. The proteins involved are frequently post-translationally modified (PTM) by ubiquitination, phosphorylation, or glycosylation to allow proper spatio-temporal regulation/recruitment of these signaling complexes in a defined cellular compartment. During the last couple of years, increasing attention has been paid to the reversible cysteine-centered PTM S-palmitoylation. This PTM regulates the hydrophobicity of soluble and membrane proteins and modulates protein:protein interaction and their interaction with distinct membrane micro-domains (i.e., lipid rafts). We conclude with which functional and mechanistic roles for S-palmitoylation as well as different forms of membrane micro-domains in death-receptor-mediated signal transduction were unraveled in the last two decades.

## 1. Introduction

The maintenance of the integrity of any multicellular eukaryotic organism is achieved by tightly balancing proliferation and differentiation vs. cell death. Regulation of the underlying processes is often mediated by activation of surface resident receptors by their respective ligands. Extrinsic apoptosis can be triggered (I) by dependence receptors, requiring a certain ligand expression threshold to avoid cell death induction and (II) death receptors, which require activation by their respective ligands [1,2].

Here, we focus on signaling pathways induced by death receptors (DR) belonging to the tumor necrosis factor receptor superfamily (TNFRSF) and their cognate ligands. In total, the TNF receptor superfamily comprises more than 27 cytokine receptors and their ligands. The subgroup of DR comprises the name-giving prototypic member TNF-R1 (CD120a, *TNFRSF1A*)/TNF-α (TNF, Cachectin, *TNFSF2*), Fas (CD95, Apo-1, *TNFRSF6*)/FasL (CD178, CD95L, *TNFSF6*), and TNF-related apoptosis-inducing ligand receptors TRAIL-R1 (CD261, DR4, Apo2, *TNFRSF10A*) and TRAIL-R2 (CD262, DR5, *TNFRSF10B*)/TRAIL (CD253, APO2L, *TNFSF10*). In addition, the less extensively studied receptors DR3 (Apo3, *TNFRSF25*)/TL1A (VEGI, *TNFSF15*), EDAR (Ectodysplasin-A receptor)/EDA protein (isoform 1), and DR6 (CD358, *TNFRSF21*), which has been reported to be activated by nAPP (amyloid precursor protein), belong to the family of death receptors. The majority of the DR are expressed ubiquitously at varying levels of abundance and can trigger diametrically opposed signaling cascades: survival, differentiation, and cell proliferation vs. various types of cell death. Activation of the differential signaling cascades is mediated by binding of the ligands to their particular receptors. TNFSF ligands are type II transmembrane proteins (intracellular N-terminus), while their receptors are type I transmembrane proteins. They are characterized by a highly conserved extracellular cysteine-rich domain (CRD), which is involved in ligand binding and self-assembly of the receptors. A hallmark of all death receptors is the cytoplasmic death domain (DD) enabling them to recruit the death-inducing signaling complex (DISC) (for review, see [3]).

For TNF-R1/TNF signaling, the predominant pathway, in most models investigated so far, is the activation of proliferative signaling. Within seconds upon ligand binding, inflammatory/proliferative signal transduction via NFκB or MAPK is activated. These initial signaling cascades emanate from plasma membrane resident receptors recruiting the *complex I*, consisting of the DD containing protein TRADD and RIP1, which are needed for TRAF2, cIAP1/2, and LUBAC (linear ubiquitin chain assembly complex) recruitment. The latter ones are required for stabilization of *complex I* signaling by K63- and M1-ubiquitination of RIP1, recruitment of TAK1/2, IKK activation, and subsequent NFκB activation. Cell death signaling via TNF-R1, on the other hand, requires K63-ubiquitination of the receptor, leading to its subsequent internalization. This allows disassembly of *complex I* in favor of *complex II*/DISC, including recruitment of FADD and caspase-8 (for review, see [3]).

For CD95 and TRAIL-R1/R2, the signaling appears to be predominantly different: FADD and Caspase-8 are directly recruited to the plasma membrane resident receptors in most (type I) cell lines, while internalization for DISC recruitment and full cell death induction is required only in type II cells. Survival signaling via NFκB occurs with a much slower kinetics compared to TNF-R1 signaling [3,4,5].

Compartmentalization of death receptor signaling from different membrane environments (plasma membrane vs. endosomes) and the cytoplasm is decisive for the respective biological outcome. One way to modulate interaction of proteins (both transmembrane and soluble) with membranes is by acylation. Peripheral membrane proteins are frequently modified at an N-terminal (MGxxxS/T) or C-terminal (CCAAx) motif by myristoylation or prenylation to increase their hydrophobicity and thus enable them to weakly interact with membranes. For augmented membrane interaction, additional acylation of cysteine residues is required. Such S-acylation also occurs in proteins containing a trans-membrane domain (TMD) [6].

Despite an increasing amount of data published in the last years, the biological outcome of protein acylation is not fully understood: soluble, hydrophilic proteins can be anchored to the cytoplasmic leaflets of cellular membranes. Depending on the type of acylation, the interaction can be weak to allow shuttling between membranes and the cytoplasm, or strong, for stable membrane interaction. S-acylation can affect protein–protein interaction and multimerization by TMD tilting and attachment of cytoplasmic protein loops to the surrounding membrane lipid environments [6].

Frequently, acylation has been reported to promote or prevent association with detergent-resistant membrane micro-domains (DRM: i.e., lipid rafts, caveolae). The existence and function of these micro-domains are highly controversial. However, it is widely accepted that the distribution of both lipids and membrane resident proteins is anisotropic and tends to form functional raft-like structures. DRM are often referred to as lipid ordered (lo) domains, whereas the surrounding area is called lipid disordered (ld). Technically, DRM are extracted from cells using weak detergents (TritonX100, Brij98) or alkaline treatment, followed by density gradient ultracentrifugation. Marker proteins for DRM are often flotillin-1 and -2, caveolin-1 and-2, or gangliosides GM1 and GM3. Caveolae can be visualized by electron microscopy, while it is hard to specifically label and detect other DRM. Frequently, cholera toxin B subunit is used to stain DRM; however, the compound has been described to trigger raft formation. Similar to labeling them, modulating/depleting DRM is unspecific and frequently affects the cells on a global level. Thus, faithful discrimination in which kind of membrane domain proteins reside is challenging [7,8,9].

Among the various lipid modifications, S-acylation is the sole reversible form. As the addition of palmitate is perceived as the predominant form (>74%) of acylation, it is usually referred to as S-palmitoylation. However, other saturated and unsaturated acyl groups could also be reversibly attached to cysteine residues [10]. S-palmitoylation is mediated at the cytoplasmic leaflet by membrane-bound palmitoyl-transferases (PAT). Whereas there is increasing knowledge about PAT:substrate interaction, little is known about their fatty acid selectivity [11]. These enzymes are mainly located at the endoplasmic reticulum (ER) and Golgi-apparatus, but also occur at the plasma membrane (PM) and on endosomes. In humans, 23 PAT exist and are characterized by their conserved Asp-His-His-Cys (DHHC)-motif. S-palmitoylation can be reverted by cytosolic palmitoyl-thioesterases (PTE). The best-described palmitoylation erasers are APT1, APT2, and PPT1, while novel classes of PTE belonging to the PSD95 and ABHD17 family of proteins have been identified in the last years [12]. Up to date, little is known about the regulation or the substrate specificity or redundancy of these enzymes [12,13]. Known functions, mechanisms, and tools to investigate S-palmitoylation are summarized in various comprehensive reviews [14,15,16].

Among other things, S-palmitoylation can be sensed as a mechanism to modulate the hydrophobic mismatch (HM) of proteins: a positive hydrophobic mismatch means the TMD of the protein is longer than the surrounding lipid environment, whereas a negative mismatch means the TMD is shorter. In general, the length of the TMDs increases from the endoplasmic reticulum resident (average length below 20 amino acids) to plasma membrane resident proteins (average length around 27 amino acids) and grows by ~1.5 Å/amino acid residue. This correlates with the thickness of the respective membranes, which are approximately 37.5 Å at the ER and 42.5 Å at the PM. Lipid rafts are supposed to be even thicker due to their high amount of sphingolipids and sterols, which can be monitored using, i.e., atomic force microscopy or cryo-electron tomography [17,18,19,20]. S-palmitoylation of proteins can for example induce tilting of TMDs having a positive HM to make them “shorter” or increase the hydrophobicity of proteins with a negative TM to allow residence in thick membranes (i.e., lipid rafts).

In this review article, we focus on the roles of membrane microdomains and palmitoylation of proteins involved in the regulation of signal transduction of the death receptor TNF-R1 and also briefly sum up similar observations for signaling via CD95, TRAIL-R1 (DR4) and –R2 (DR5), and DR6.

## 2. Roles of Palmitoylation and Lipid Rafts in TNF-R1 Signaling

TNF is a type II single spanning membrane protein that can activate both TNF-R1 and TNF-R2. While soluble TNF (sTNF) has a higher affinity for TNF-R1, membrane TNF (mTNF) stimulates both receptors [21]. Conversion of mTNF to sTNF is mediated by ADAM17/TACE [22,23]. TNF palmitoylation at cysteine 47 has been described first in COS-1 (green monkey kidney fibroblast cells) and Sf9 insect cells [24]. This observation was confirmed in two other studies, additionally showing that palmitoylation of TNF regulates TNF lipid raft association and sTNF vs. mTNF activation of TNF-R1 in HeLa (human cervix), Raji (human B lymphocytes), and T24 (human urinary bladder) as well as in murine 3T3L1 fibroblasts and RAW264.7 macrophage cells [25,26]. In human monocytic U937 cells, sTNF activates NFκB signaling at plasma membrane resident receptors, followed by receptor internalization and DISC formation. mTNF triggers internalization-independent DISC formation at PM resident TNF-R1 in a STAT1-dependent manner [27].

We described palmitoylation of TNF-R1 itself in U937 cells for the first time [28]. TNF-R1 is at least double palmitoylated (Figure 1A). C248 palmitoylation is required for receptor translocation to the cell surface (Figure 2A). The functional consequence of palmitoylation of other cysteine residues and which residues are modified remains to be investigated. Requirement of C248 palmitoylation for PM transport may explain why the majority of TNF-R1 resides in the Golgi-apparatus [29,30]: the TNF-R1 TMD comprises 21 residues fitting to thin membranes (i.e., ER and Golgi), whereas it has a negative HM for PM membrane micro-domains. Lipid modification ameliorates this mismatch and enables transport to the cell surface.

TNF-R2, which is the second TNF binding receptor (lacking a death domain), contains a longer TMD (30 amino acid residues) and is predominantly located at the plasma membrane [31]. TNF-R2 contains one cysteine residue within its TMD and five more in its cytoplasmic tail, thus making it an ideal target for palmitoylation (Figure 1B). However, experimental validation is lacking.

Constitutive TNF-R2:TRAF-2:caveolin-1 interaction in the same protein complex enables it for NFκB activation in human umbilical vein endothelial cells (HUVEC). Constitutive TNF-R1:caveolin-1 interaction could not be observed in the same cell line, supporting later observations that TNF triggers TNF-R1 lipid raft translocation prior to NFκB activation [32]. As described for TNF, TNF-R2 is also shed by ADAM17 in lipid rafts of leukocytes [33].

In untreated U937 and ECV304 (genetically identical with T24/83 cell line [34]) cells, the majority of TNF-R1 protein is located in the Golgi-apparatus [29,30]. Deletion of the TNF-R1 intracellular domain results in lost Golgi but increased PM localization in HUVEC cells; however, fusion of TNF-R2 with the TNF-R1 intracellular domain did not restrict the chimeric protein to the Golgi. This can be explained by the long TNF-R2 TMD. In contrast, it was shown that DD deletion results in reduced lipid raft localization of TNF-R1 and a uniform distribution in the plasma membrane of HeLa cells [35]. Such DD deletion would result in lacking palmitoylation of a cysteine residue in this area (C395 or C433) (Figure 1A).

The palmitoyl transferase(s) required for TNF-R1 palmitoylation remains enigmatic. However, upon TNF stimulation, APT2 (LYPLA2) triggered partial de-palmitoylation of TNF-R1. This was required for NFκB activation. Using PTE-selective fluorescent probes, we observed activation of APT2 in response to TNF, while it is still unclear how APT2 is activated and recruited to TNF-R1. Unpublished data of our group showed that APT2 activity facilitates lipid raft translocation of TNF-R1. This process may require additional palmitoylation at hitherto unknown cysteine residue(s) to allow receptor localization in micro domains. Our observation is in line with another report showing that NFκB activation requires lipid raft translocation in HT1080 fibrosarcoma cells [36] (Figure 2B). In contrast, in a rat model, it was shown that TNF-induced apoptosis but not NFκB activation was associated with lipid rafts: Receptor activation triggered TNF-R1 but not TNF-R2 or CD95 lipid raft translocation. Lipid raft translocation also involved TNF-R1 and TRAF2 ubiquitination [37]. In their study, Legler et al. also observed poly-ubiquitination of TNF-R1 and RIP1 upon lipid raft translocation. We observed that TNF-R1-mediated apoptosis requires K63-ubiquitination and subsequent internalization of TNF-R1 [38] (Figure 2C). In line, HRG (histidine-rich glycoprotein) interaction with TNF-R1 results in TNF-R1 K63-ubiquitination and apoptosis induction [39]. HRG has been described to be multiply palmitoylated in murine cells [40]. A report by the Walczak group showed that M1-ubiquitination of TNF-R1 and RIPK1 is required for NFκB signaling [41]. Thus, we hypothesize that differential TNF-R1 M1/K63-ubiquitination occurs in distinct membrane environments (i.e., lipid rafts for NFκB signaling via *complex I* and liquid disordered (ld) membranes or caveolae for endocytosis). This is regulated by the palmitoylation status of the receptor and, putatively, other proteins (i.e., E3 ubiquitin ligases).

Intracellular bacteria and also viruses can inhibit TNF-R1 internalization to prevent apoptotic killing of the host cells and thereby evade elimination by the host immune system. It is not known if these processes also involve palmitoylation or alterations in membrane lipid composition [42,43,44,45,46]. The *P. aeruginosa* quorum-sensing molecule 3oc, however, interacts with the PM of human cells. This triggers spontaneous TNF-R1 multimerization in liquid disordered (ld)/non-raft phase and apoptosis induction in various cell lines. It is not clear if this also affects receptor internalization and ubiquitination [47].

Connections between TNF-R1 signaling and detergent-resistant membranes/lipid rafts/caveolae have been reported by various groups. Association of TNF-R1 with caveolae had first been described in 1999, showing that TNF-mediated apoptosis requires caveolae and ceramide production by neutral sphingomyelinase (nSMase) in U937 cells (Figure 2D). CD95-mediated cell apoptosis, however, did not require caveolae either in U937 or in Jurkat T-lymphocytes [48]. This supports our data, showing that upon pharmacological inhibition of APT2, TNF-R1 stays at the PM (putatively in caveolae) and induces ceramide production via nSMase in U937 cells. The Yates group showed that TNF-R1, CD95, and DR5 are located in caveolin-1 positive DRM fractions of U-1242 glioma cells by default and that TRAIL treatment invoked caspase-8 recruitment and activation in this compartment [49,50]. D’Alessio et al. suggested a TNF-R1-membrane-proximal sequence, close to C248, as we recently reported, targeting it to caveolae and that TNF-R1 internalization requires caveolae localization [51].

However, a discrepancy exists regarding the mode of TNF-R1 internalization. Currently, endocytotic mechanisms are classified into three types: (a) non-lipid-raft-dependent (clathrin-mediated endocytosis: CME), (b) lipid-raft-dependent (caveolae-mediated: CM, flotillin-dependent: FD), and (c) mixed-membrane-dependent (phagocytosis and micropinocytosis). TNF-R1 but not CD95, DR4, or DR5 contains an YXXΦ internalization motif (TRID, TNF-R1 internalization domain), enabling the receptor to interact with the CME machinery.

Several reports showed that TNF-R1 requires CME for apoptosis induction (Figure 2C). This step could be blocked pharmacologically (using Monodansylcadaverine, Dynasore, or Pitstop) or by mutagenesis of the TRID [27,38,52,53,54,55,56,57]. Adenovirus infection as well as the expression of the adenoviral protein E3-14.7K results in inhibition of TNF-R1 internalization, serving as an immune escape mechanism [46]. In addition, TNF-R1 internalization and cell death induction is blocked in CerS2^-/-^ mice as well as in cells lacking ST6Gal-I sialyltransferase [56,57,58]. Sialylation of CD95 by ST6Gal-I also blocked FasL-mediated cell death, while DR4 and DR5 were not affected likewise in HD3 colon carcinoma cells [59]. The requirement of receptor internalization for signal transduction differs among DR’s as well as of the respective investigated model. Details are given in the respective paragraphs.

As stated above, the previous reports regarding translocation of TNF-R1 to lipid rafts support our own observation that NFκB induction required receptor de-palmitoylation by APT2 and lipid raft translocation. On the other hand, inhibition of APT2 activity resulted in massive neutral sphingomyelinase (nSMase)-dependent ceramide production and cell death induction [28]. TNF-mediated nSMase activation, ceramide production, and finally cytotoxicity are also required for dopaminergic neuron formation [60]. At the moment, it is not clear which nSMase isoform is responsible for this effect. The human genome encodes four different nSMase isoforms, including nSMase1 (*SMPD2*), nSMase2 (*SMPD3*), nSMase3 (*SMPD4*), and the mitochondrial ma-nSMase (*SMPD5*) [61]. nSMase2 is activated by TNF via PKC-δ and MAPK and is directly linked to TNF-R1 via FAN and RACK1 [62,63,64,65]. Similar to our observation that APT2-inhibition results in ceramide formation and apoptosis induction without further affecting receptor endocytosis, TNF-R1 internalization deficient cells trigger cell death via nSMase-mediated ceramide formation [66]. In neurons, TNF-dependent formation of cytotoxic ceramide has been described, while pharmacological nSMase inhibition dampened TNF-induced apoptosis [60]. Caveolin-dependent activation of nSMase as well as cytotoxic ROS formation upon TNF stimulation has been reported in LHCN-M2 human muscle satellite cells [67]. ROS production upon TNF stimulation has also been described in HeLa and MEF cells; however, in contrast, recruitment of the NADPH complex occurred in TNF-receptosomes [68].

nSMase2 palmitoylation has no effect on enzyme activity but affects its membrane association [69,70]. Co-localization of both nSMase2 and palmitoyl protein thioesterase 1 (PPT-1) has been observed in lipid rafts. nSMase2 overexpression enhanced staurosporine and C2-ceramide-induced cell death, while the opposite was observed for PPT-1 overexpression [71].

PPT-1 is involved in TNF-R1-mediated apoptosis induction; however, whether it modulates TRAIL or FasL-induced signaling is not clear. PPT-1 deficient fibroblasts from patients suffering from infantile neuronal ceroid lipofuscinosis (INCL) or murine cells harboring a disrupted *Ppt1/Cln1* gene revealed reduced caspase activation, Bid-cleavage, and cytochrome C release by mitochondrial outer membrane permeabilization (MOMP). NFκB activation and MAPK signaling were unaffected. Apoptosis induction via staurosporine was not affected [72]. Possible substrates that are located outside the lysosome and involved in TNF (or other DR) triggered cell death are not known.

Recently, we described HSP90β as a proteolytic substrate of the lysosomal aspartic protease Cathepsin D (CtsD). A distinct amount of cellular HSP90β is cleaved as a response to TNF and TRAIL. Our data suggest that a small fraction of HSP90β is also palmitoylated and encounters CtsD upon its release from lysosomes at the PM. However, this observation requires further investigation [73]. Palmitoylation of HSP90 has been shown in murine cells [74]. One hypothesis might be that the palmitoylation of HSP90β or even CtsD may be linked to PPT-1 activity. Intriguingly, in PPT1 knockout mouse models for INCL, increased CtsD expression has been reported [75,76,77].

PPT-1 was described as being palmitoylated either via zDHHC3 or 7; however, mutation of the palmitoylation site did not alter subcellular localization, but rather affected its de-palmitoylation activity [78]. Whether PPT-1 or zDHHC3/7 are involved in nSMase2 palmitoylation or not is speculative. We observed increased levels of palmitoylated PPT-1 in response to TNF in U937 cells [28].

nSMase3 activation upon TNF stimulation has been observed in MCF7, human mammary gland, cells. Unlike nSMase2, nSMase3 is linked to membranes via its C-tail [79]. TNF stimulation of murine muscle cells resulted in ceramide production by nSMase3, which co-fractionated with detergent-resistant membranes [80].

TNF-R1 internalization prior to apoptosis induction and γ-secretase-mediated release of *complex IIa* bound to a TNF-R1 ICD (intracellular domain) has been described in MCF7 cells [52]. Traditionally, it is assumed that ectodomain shedding by ADAM proteases precedes intracellular cleavage by γ-secretase (Figure 2E). However, in HUVEC and Cos7 cells, extracellular aSMase activates ADAM17, leading to TNF-R1 shedding and reduced apoptosis, as TNF cannot bind the receptor [81]. Shedding of TNF, TNF-R1, and TNF-R2 by ADAM17 as well as association of the shedding activity with caveolin-1 positive lipid rafts of monocytic THP-1 and EA.hy926, umbilical vein cells, had been described before [82,83].

In contrast to the above-mentioned reports, TNF-R1 was shown to be endocytosed from caveolae in a clathrin-independent manner, and lipid raft depletion using β-methyl cyclodextrin (βMCD) did not alter IκBα degradation/NFκB activation. Phosphorylation of Akt and thus protection from TNF-mediated cytotoxicity was inhibited in EA.hy926 cells [84]. Primary murine macrophages have been used to show that TNF triggers lipid raft localization of TNF-R1. Interestingly, βMCD treatment abrogated TNF-induced MAPK but not NFκB signaling [85]. Additionally, TNF triggered TNF-R1 translocation towards lipid rafts in airway smooth muscle cells. However, depletion of lipid rafts using βMCD did not affect NFκB and MAPK signaling, while RhoA GTPase signaling was blocked in this cell type [86]. Interestingly, RhoA GTPase levels are RNF8-dependent and regulate metastasis in triple-negative breast cancer [87].

TNF-induced necroptosis induction was also linked to lipid rafts. Necroptosis is activated in favor of apoptosis in cells lacking caspase-8 enzyme activity, resulting in necrosome formation instead of *complex IIa* or *IIb* (Figure 2G). Two studies revealed that treatment of cells with Phenhydan^®^ blocked TNF-R1-induced NFκB activation by affecting lipid raft association. Phenhydan^®^-treatment also blocked necrosome formation in both studies [88,89]. Mechanistically, Phenhydan^®^-treatment resulted in disperse TNF-R1 surface distribution by disrupting plasma membrane organization [89]. Interestingly, TNF-induced necroptosis induction was reduced in fibroblasts derived from ADAM17^ex/ex^ mice, suggesting that ADAM17 may be involved in redirecting signaling between apoptosis and necroptosis at least in mice. Our own unpublished data showed that pharmacological inhibition of ADAM17 or ADAM10 neither boosted nor inhibited apoptosis or necroptosis in cells of myeloid lineage.

Ali and colleagues reported that the TNF-R1-dependent necrosome formation occurs at caveolin-1-positive intracellular organelles, a process that is blocked upon herpes simplex virus infection as an immune escape mechanism [90]. Intriguingly, necrosome-induced damage at the PM appears to be salvaged by capturing phosphorylated MLKL in Flotillin-1/-2 and ALIX positive organelles before the cell ruptures [91] (Figure 2H). The requirement of ALIX-palmitoylation for exosome formation has been reported and is likely involved in protection from necroptosis [92]. Experimental evidence, however, is lacking.

Pyroptosis is another form of necrotic cell death that can be triggered by chemotherapy, but also by TNF. Caspase-3 cleaves gasdermin E (GSDME), which results in membrane perforation and cell death. GSDME is palmitoylated at C407/408 in response to TNF stimulation in HCT116 (human colon carcinoma) and HeLa cells [93,94].

The above-described findings regarding roles of palmitoylation and liquid-disordered (ld)/liquid-ordered (lo) membranes in the subcellular compartmentalization of TNF-R1-mediated signal transduction are schematically depicted in Figure 2.

## 3. Roles of Palmitoylation and Lipid Rafts in CD95 Signaling

As described for TNF-R1, CD95 also relays both cell death and survival signals. Initially, it has been thought that the CD95 system has its main role in the maintenance of immune homeostasis and tumor elimination via apoptosis induction. During recent years, however, it has become clear that CD95 also promotes inflammation and carcinogenesis by activating non-apoptotic signaling [5,95].

Palmitoylation of FasL (CD95L) has been reported by Guardiola-Serrano and colleagues: they showed that palmitoylated FasL associates with lipid rafts in T-cells. Here, it interacts with its receptor and is shed by ADAM10 protease for cell death induction in the target cell. It was described that only palmitoylated FasL can trigger cell death [96]. Within lipid rafts of secretory lysosomes from the T-cell, the Fas ligand interacts with both ADAM10 and its close relative ADAM17. This interaction can be affected by cholesterol depletion. TCR-activation resulted in ADAM17-mediated FasL shedding in these organelles [97].

CD95 palmitoylation at cysteine 199 in human and cysteine 194 in murine CD95 is required for association with cytoskeleton-linked lipid rafts, formation of micro-aggregates, and full cell death induction in human embryonic kidney (HEK293), SKW6.4 (human B lymphocytes), H9 (human T lymphocytes), and murine NIH3T3 cells [98,99] (Figure 1C). DISC formation upon CD95 activation occurs in lipid rafts in both type I and type II cells [100]. This supports the observation that FasL:CD95 interaction and cell death induction occurs in lipid rafts of Jurkat cells [96]. A palmitoylation deficient CD95 variant (C194V) prevented receptor-mediated apoptosis induction in murine primary T-, B-, and dendritic cells [101].

CD95 palmitoylation by zDHHC7 is required for its stability by preventing lysosomal degradation and facilitating transport to the plasma membrane of colorectal cancer cells [102]. Interestingly, zDHHC7 together with zDHHC21 mediates palmitoylation of the DRM protein caveolin-1 in rat hippocampal neurons as well as in HEK293 cells. The cooperation of these proteins may thus also affect CD95 or other DR signaling [103].

The CD95 TMD (18 amino acid residues) prefers a liquid-disordered (ld)/non-raft environment, due to the negative hydrophobic mismatch between the length of hydrophobic TM domain and the thickness of liquid-ordered (lo) membranes. It was shown that ceramide does not trigger translocation of CD95 to lipid rafts but rather traps CD95 in lipid rafts due to reduced lateral diffusion in an artificial membrane system [104]. Haynes and colleagues showed that CD95 localization can differ in Burkitt’s lymphoma (BL) cells. In group III BL cells, CD95 can be detected both in the Golgi and the PM, whereas Golgi localization of CD95 dominates in group I BL cells. In the latter, CD95 localization can be triggered in favor of PM localization by CD40 ligation [105]. However, whether this process involves palmitoylation remains to be investigated.

The association of CD95 with DRM and a cell-line-dependent requirement for DRM in cell death induction has been shown in a number of reports. CD95 induced cell death in human retinal pigment epithelial cells (ARPE-19), which required GM1-positive lipid raft localization [106]. CD95 clustering in lipid rafts was also required for ROS production in CD95 enriched lipid rafts in bovine coronary arterial endothelial cells (CAEC) [107,108]. Presumably, this process also involves fusion of NADPH oxidase containing vesicles with the PM, a mechanism that had been already described previously [109]. Additionally, reactive oxygen species are required for translocation of aSMase to the PM [110].

The alkyl-lysophospholipids Edelfosine and Perifosine and the didemnin Aplidin appear to mimic ceramide and trigger CD95 localization in lipid rafts, resulting in cell death by DISC recruitment even in the absence of FasL in Jurkat cells [111,112,113,114]. This ligand-independent mechanism has been termed CASMER (clusters of apoptotic signaling molecule-enriched rafts) formation [115].

Cisplatin induces enhanced aSMase-dependent clustering of CD95 in lipid rafts in HT29 cells, derived from the human colon, which sensitizes the cells for ligand-induced cell death [116]. Resveratrol similarly triggered CD95, as well as DR4 and DR5, clustering in lipid rafts. Here, DISC formation occurred either requiring additional receptor activation via agonistic Ab-CH11 or even independent of ligand binding [117,118]. Likewise, Resveratrol induces DISC formation followed by MOMP, independent of CD95 activation in Jurkat and multiple myeloma cells [119]. Lipid raft clustering of CD95 followed by cell death induction is obviously a common way to sensitize cells to be killed by various compounds. Docosahexaenoic acid, TSWU-BR23, and endocannabinoids were shown to enhance receptor-mediated cell death in various experimental settings connected to receptor translocation into lipid rafts [120,121,122]. Cryptocaryone treatment results in CD95, FADD, and Caspase-8 and also DR4 and DR5 clustering in lipid rafts. However, cryptocaryone also induced clustering of these molecules in PC-3 prostate adenocarcinoma cells pre-treated with βMCD for lipid raft depletion [123]. Overall reduction of lipid rafts by the fluoropyrimidine drug-candidate F10 resulted in CD95 activation. This was due to enrichment and co-localization of both CD95 and CD95L in the remaining lipid rafts [124].

The opposite has been observed for CD95-mediated cell death: Ko and colleagues showed that CD95-mediated killing does not require caveolae in Jurkat and U937 cells [48]. Lipid raft disruption by ginsenoside Rh2 induced CD95 oligomerization and cell death in HeLa cells [125]. A similar observation has been made in HaCaT human keratinocyte cells, revealing that depletion of lipid rafts by various compounds induced CD95 clustering and DISC formation outside rafts [126]. As for TNF signaling, there are obviously cell-line- and organism-specific regulatory mechanisms, which may be due to differential lipid composition.

Furthermore, protein interactions can enhance CD95 recruitment to lipid rafts. Semaphorin-3A (Sema3A) and its receptor plexin-A directs CD95 to lipid rafts. This requires actin-cytoskeleton remodeling in *type I* but not *type II* cells [127]. As Sema3A is constitutively expressed in CD4/CD8 thymocytes, it could have a role in T-cell homeostasis [128], as for elimination of effector memory T-cells depend on TCR (T-cell receptor) and CD95 clustering in lipid rafts [129]. Additionally, TCR-re-stimulation of CD4^+^-T-Cells resulted in CD95 recruitment to lipid rafts in the absence of ligand binding [130].

Besides proteins and various chemical compounds, irradiation with UV light has been reported to induce clustering of CD95 in lipid rafts. Depending on the cell line, different modes of cell death could be induced with or without additional ligand binding [131,132]. UV light induces CD95- but not TNF-R1-dependent aSMase activation in MCF7 cells, which might be required for translocation to lipid rafts [133], a similar effect that had been reported in HT29 colon cancer cells [116]. Besides CD95 ligation, UV-C radiation also induces aSMase translocation into rafts in Jurkat cells. Unlike for CD95 activation, this does not involve caspase activity [134].

Roles for aSMase and sphingolipids for CD95 clustering have been reported in several studies before: pharmacological inhibition of sphingolipid synthesis by myriocin reduced CD95-mediated cell death in T-cells [135]. Several groups reported that sequential activation of CD95 in distinct membrane compartments is required for cell death induction.

In sum, signaling via CD95 appears to follow the schemes depicted in Figure 3. In *type I* cell lines, DISC formation occurs rapidly at the PM and is sufficient for apoptosis induction. *Type II* cells require the activation of low amounts of caspase-8 at the PM, triggering the formation of SPOTS (signaling protein oligomerization transduction structures) and internalization of the activated receptors. CD95-receptosomes undergo intracellular maturation and trigger LMP and MOMP for apoptosis amplification [109,136,137,138,139,140]. aSMase is transported to the PM via fusion with trans-Golgi or lysosomal vesicles [108,109,110], where it is involved in the regulation of lipid raft size and PKCδ recruitment, linking rafts to the cytoskeletal machinery [141].

The role of other post-translational modifications of CD95 has been summarized in a comprehensive review [142].

## 4. Roles of Palmitoylation and Lipid Rafts in TRAIL-R Signaling

Differently from TNF and FasL, palmitoylation of TRAIL has not been reported. However, its cytoplasmic part contains a possible palmitoylation acceptor site at Cys16, which is close to its TMD. Using receptor-selective agonistic antibodies as surrogates for TRAIL, Wajant and colleagues reported that DR4 can relay cell death via both soluble (sTRAIL) and membrane TRAIL (mTRAIL), whereas DR5-mediated death can only be induced by mTRAIL [143]. However, whether the sheddase is involved in sTRAIL release and lipid rafts regulate TRAIL shedding is enigmatic.

Palmitoylation of TRAIL-R1 (DR4) at membrane-proximal cysteine residues 261/262/263 is critical for DR4 lipid raft localization, oligomerization, and consequently, cell death induction [144]. Besides the described cysteine residues, DR4 comprises four more potential palmitoylation sites (Cys 268, 274, 279, and 336); none of these sites have been reported to be palmitoylated (Figure 1D). The TRAIL-R2 (DR5) amino acid sequence contains three putative palmitoylation sites (Figure 1E). The protein has not been reported to be palmitoylated so far.

However, similar to TNF-R1 and CD95 signaling, the association of TRAIL-Rs with lipid rafts has been reported in several studies in various cell lines and conditions. Cdc42-associated kinase 1 (Ack1) regulates DR4 but not DR5 oligomerization and lipid raft localization, and cell death induction in MCF10A breast epithelial, but not NCI-H460 lung carcinoma cells [145]. Shedding of DR4 by TACE as a possible mechanism for de-sensitization has been reported in myeloma cells. However, DR5 was not shed. Whether shedding depends on lipid raft localization is not clear but could be assumed depending on the observations from the CD95 and TNF-R1 systems [146].

Recruitment of DR4 and/or DR5 to lipid rafts can be induced by various chemical compounds without previous receptor activation: DR5, CD95, and TNF-R1 clustering in lipid rafts is enhanced in Jurkat cells upon treatment with Aplidin and thus sensitization for cell death. The mechanism is cytoskeleton-dependent [111]. In Jurkat cells, DR5 associates constitutively with GM1 positive lipid raft fraction, while FADD, caspase-8, and PI3K-p58, which is involved in Golgi-vesicle/endosomal trafficking, were recruited upon stimulation with sTRAIL [147].

Treatment of multiple myeloma B cells with synthetic alkyl-lysophospholipids results in accumulation of DR4, DR5, and CD95 as well as Bid in lipid rafts, facilitating cell death induction [113]. Localization of DR4 in GM3-positive lipid raft fractions as a prerequisite for cell death induction has been described in B-cells [148]. DR4-mediated cell death was also observed in some chronic lymphocytic leukemia cell lines, while other cell lines required additional fludarabine treatment, triggering DR4 but not DR5 lipid raft localization [149].

In non-small-cell lung carcinoma cells (NSCLC), lipid rafts appear to be required for TRAIL-mediated cell death induction, while NFκB and MAPK signaling occurs outside rafts. In TRAIL-sensitive cells, DISC assembly can occur in both rafts and non-rafts, whereas in TRAIL-resistant cells, DISC assembly occurs only outside rafts [150]. Similar, DR4 and DR5 aggregated in lipid rafts in TRAIL-sensitive but not resistant NSCLC [151].

In glioblastoma cells predominantly expressing DR5, the binding of TRAIL resulted in DISC formation in lipid rafts, whereas NFκB activation occurred outside lipid rafts [152]. Temozolomide caused DR5 accumulation in lipid rafts of U251 glioma cells and thereby triggered cell death [153].

In gastric cancer cells, the expression of DR5 could be enhanced upon stimulation with Ursodeoxycholic acid (UDCA). Such treatment also triggered DR5-containing lipid raft formation and ROS production in lipid rafts [154]. In line with this, TRAIL-resistant MGC803 gastric carcinoma cells could be sensitized for cell death by incubation with epirubicin. Epirubicin alone triggered DR4 and DR5 aggregation in lipid rafts and ligand-independent cell death induction. Treatment of TRAIL together with epirubicin enhanced TRAIL-induced cell death. However, this effect could be partially reverted by lipid raft depletion using nystatin [155].

In TRAIL-resistant gastric cancer cells, TRAIL induces translocation of EGF-R to lipid rafts, resulting in reduced DISC formation and thus inhibition of cell death. In lipid rafts, EGF-R competes with TRAIL-R’s for Cbl-b, which is crucial for DISC formation [156]. TRAIL also enhances caveolin-1 activation and Src tyrosine kinase translocation to lipid rafts. Inhibition of Src enhanced TRAIL-mediated cell death by inhibition of Src-EGF-R and Src-caveolin-1 interaction [157]. The same group reported that TRAIL enhances IGF-1R expression, Cbl-b-dependent recruitment to lipid rafts, and IGF-1R-mediated anti-apoptotic signaling [158].

In murine cells, TRAIL activated aSMase (Smpd1) and Lysosome/PM fusion, resulting in GM1 positive lipid raft formation, DR4 clustering, and NADPH oxidase activation [159].

Apart from signaling from the plasma membrane, roles of TRAIL-Rs and to a lesser extend also TNF-R1 in endoplasmic reticulum (ER) stress response resulting in both cell death and NFκB signaling have been described. As the ER is a major hub for lipid metabolism, one could assume that ER-lipid rafts also modulate TRAIL-R-mediated stress response by CASMER formation at the ER. However, to date, no such link has been established [160,161].

## 5. Palmitoylation in DR6 Signaling

DR6 is another member of the TNF-R superfamily involved in neuro-development, but it also regulates tumor extravasation and metastases [162]. Besides being N- and O-glycosylated, DR6 is palmitoylated at membrane-proximal Cys368 [163] (Figure 1F). In the case of DR6, palmitoylation appears not to be required for lipid raft localization.

The sole known cell-death-inducing DR6 ligand is APP (amyloid precursor protein). The membrane-bound form of APP is required to induce cell death [162,164,165]. Palmitoylation of APP occurs at Cys186 and Cys187. Cleavage of APP by Beta-secretase 1 (BACE1) promotes its dimerization and palmitoylation [166,167]. APP internalization mainly occurs in lipid raft micro domains [168]. If and how this affects DR6 signaling is not clear.

## 6. Palmitoylation and Lipid Raft Association of Other Proteins in Death-Receptor-Mediated Signal Transduction

The Bcl-2 protein family member Bax, another pro-apoptotic protein, has been described as being palmitoylated (Cys62 and 126). Whether Bax palmitoylation is required for MOMP induction and if such modification occurs in response to TNF, TRAIL, or FasL ligation is not yet clear [169]. MOMP and cytochrome C release involves Bax interaction with ceramide-rich macro-domains in the outer mitochondrial membrane of HeLa cells upon radiation [170]. One Bax-interacting protein is the Bcl-2 protein Bid, which is cleaved in response to TNF-R1 activation by caspases or Cathepsin D. The resulting p15 Bid fragment is myristoylated at its N-terminus, which is required for MOMP and cytochrome C release [171,172,173,174,175]. Sorice and colleagues reported that CD95 activation induces the translocation of the ganglioside GM3 from the PM towards the outer mitochondrial membrane. Here, GM3 is involved in MOMP. In line with these experiments, they also reported that CD95 activation triggers translocation of cellular prion protein (PrP^c^) to mitochondrial lipid micro-domains of T cells, where it is involved in cell death regulation [176,177,178,179,180].

Caspase-6, which is involved in pathogenesis of neuronal diseases, has been described to be palmitoylated by zDHHC17/HIP14 twice, at Cys264 and 277. Palmitoylation of caspase-6 reduces its activation [181]. Besides murine caspase-6, human caspase-8 and murine caspases-1 and -3 have been identified as putatively palmitoylated in palmitoyl-proteomes, based on the Swiss Palm database; however, further experimental evidence is lacking [182]. Compared to other caspases, caspase-6 is poorly characterized [183]. Zheng and colleagues recently identified caspase-6 as a regulator of inflammation and immune signaling against influenza A virus infection [184]. However, whether the palmitoylation state of caspase-6 also regulates DR signaling is unknown. Pointing in this direction is the fact that in Jurkat cells, CD95-dependent apoptosis induction involves nSMase-mediated ceramide formation and caspase-6 activation in the nucleus [185].

Pro-caspase-8 recruitment to activated TNF-receptosomes requires the ESCRT (endosomal sorting complex required for transport) proteins ALIX and ALG-1 [186,187]. Non-canonical DISC formation, i.e., caspase-8 activation on auto-phagosomal membranes, has been described as being dependent on the ESCRT protein CHMP2A [188]. Interaction of ALIX with TNF-receptosomes putatively occurs via its K63-ubiquitin binding domain and K63-ubiquitinated TNF-R1; however, final experimental evidence is lacking [38,189]. Palmitoylation of ALIX has recently been described and is required for CD9 interaction and exosome formation [92]. However, whether ALIX’s palmitoylation state also regulates TNF-signaling/trafficking or prevention of pMLKL-induced PM damage, as recently described, appears likely but is not known [91].

Tyrosine kinases regulate T-cell function, i.e., by modulating DR-mediated killing [190]. Lck requires palmitoylation by zDHHC21 and lipid raft association to allow CD95-mediated killing in Jurkat cells, which involves palm-Lck-dependent PLC-γ1 (phospholipase C γ1) activation [191]. Lck also regulates TRAIL signaling in T-cells. TRAIL binding of DR4-Fc together with CD3 activation results in recruitment of Lck to lipid rafts and NFκB activation [192]. Besides Lck, the subcellular localization of other Src-family tyrosine kinases also depends on their palmitoylation state [193]. Fyn is palmitoylated and interacts with zDHHC5. It phosphorylates flotillins, which in turn are palmitoylated by zDHHC5, are recruited to lipid raft micro domains, and induce flotillin-dependent internalization of surface molecules [194,195].

When writing about lipid rafts, ceramide synthases (CerS) and their roles in DR signaling cannot be ignored. CerSs regulate the length and saturation of fatty acids in ceramide [196]. Together with the Futerman group, we showed that in CerS2 knock out mice, TNF-mediated cell death is abrogated. This is due to lacking TNF-R1 internalization [58]. Similarly, knockdown of CerS6 in SW480 colon adenocarcinoma cells reduced TRAIL-mediated apoptosis and CerS6 overexpression enhanced TRAIL-mediated cell death in SW620 colon carcinoma cells [197]. In contrast, CerS6 overexpression confers resistance to CD95-mediated cell death by inhibition of DISC formation at the receptor [198].

## 7. Death Receptor Signaling from the Nucleus

Signaling of plasma membrane-localized receptors in the nucleus has been frequently reported for members of the receptor tyrosine kinase family. Often, this involves the release of an intracellular domain (ICD) upon extracellular α- and intracellular γ-secretase cleavage. However, full-length protein has also been found to be localized in the nucleus upon ligand binding [199]. During the last years, evidence for roles of DR4 and DR5 in the nucleus is accumulating. In all cases, nuclear TRAIL-R localization has been reported to promote cell survival [200]. Recently, Mert and colleagues showed that both receptors can be endocytosed upon TRAIL binding by CME from the PM and undergo trafficking to the nucleus [201]. However, it is not clear how full-length DR4 and DR5 can enter the nucleus and exert their function in this compartment. It is also enigmatic if this process involves DRM or protein palmitoylation. Likewise, it is not known if and how endogenous nuclear-localized TRAIL may affect signaling. Nuclear localization of TNF-R1, CD95, or DR6 has not been reported up to now. For TNF-R1, intracellular cleavage by γ-secretase suggests ICD generation in some cell lines; however, no nuclear localization or function of this ICD has been reported yet [52]. Localization of the DR3 ligand TL1A in the nucleus but not the receptor has been described in psoriatic skin [202]. While little is known about the roles of PTM of DR3 in its signaling cascades, its similarity to TNF-R1 suggests similar regulation (Figure 1G). Nuclear localization of the TNF-R superfamily member CD40 has also been reported in human B lymphocytes; however, CD40L was not present in the nucleus [203]. CD40 activates NFκB signaling and constitutively resides in lipid rafts [204]. CD40 contains one cytosolic cysteine residue (C258 = C238 without signal peptide), representing a possible palmitoylation site; however, as it is 42 amino acids away from the TMD, it is unlikely to regulate lipid raft association (Figure 1H). C258 of CD40 has rather been described to be required for receptor dimerization by cysteine bridge formation [205]. However, bridge formation and palmitoylation may compete to regulate CD40 signaling.

## 8. Myristoylation in Cell Death

Frequently, soluble proteins are modified at an N-terminal (MGxxxS/T) motif by myristoylation. This increases their hydrophobicity and thereby enables them to interact weakly and transiently with membranes. Myristoylation had been thought to mainly occur co-translationally and is mediated by the *N*-myristoyltransferases NMT1 and NMT2. However, several studies showed that cleavage of proteins by caspases upon cell death induction liberates novel N-terminal myristoylation acceptor sites in the respective target proteins. Several proteins involved in the regulation of T cell death have been described to be affected likewise: upon caspase-8 cleavage, tBID is myristoylated, allowing its recruitment to mitochondria. Similarly, actin and gelsolin are cleaved and then myristoylated. Myr-actin has pro-apoptotic activity, while myr-gelsolin has anti-apoptotic activity [206,207]. PKCε is another substrate for myristoylation upon caspase cleavage [208]. PKCε regulates TNF-R1 shedding upon Bryostatin-1 treatment of T84 cells, concomitant with increased TNF-R1:PKCε interaction [209]. In U937, Jurkat, and K562 bone-marrow cells, TNF triggers PKC translocation to the plasma membrane [210]. PKC activity had been shown to block NFκB activation via TNF-R1 as well as *complex II* formation at DR4 and DR5 in HeLa cells [211]. Src kinase activity is also regulated by myristoylation [212]. Src-mediated Y380 phosphorylation of caspase-8, prevents caspase activation and thereby, dampens CD95-mediated cell [213].

Interestingly, both NMT1 and 2 are substrates for caspase-3 and -8. Cleavage results in their subcellular re-localization without alteration in enzyme activity [214].

In none of these studies, cell death was induced via death receptors. However, it is tempting to speculate that DR-mediated cell death also triggers myristoylation of these and other proteins.

## 9. Alterations in the Lipid Composition of Membranes—Opportunities for Clinical Exploitation

Interfering with the lipid metabolism of a cell, tissue, or organism will affect their response towards a plethora of stimuli (i.e., cytokines or pathogens). Altering lipid metabolism changes the ability of a cell to interact with its environment by altering its membrane composition as well as the availability of lipids for protein acylation. This can be therapeutically exploited. In clinical treatment of various diseases ranging from cancer to auto-immune diseases and infection, statins (i.e., lovastatin or simvastatin) are routinely applied as inhibitors of cholesterol biosynthesis. Statin application affects both cholesterol levels in cellular membranes as well as in the blood stream and consequently affects cellular mechanisms and signaling pathways relying on membrane compartmentalization [215,216].

Among the hallmarks of cancer is the ability for sustained angiogenesis, invasion, and metastasis, which are highly associated with altered lipid composition and membrane fluidity of cancer cells. Approaches to targeting lipid rafts in cancer cells have been summarized in various comprehensive reviews [217,218,219].

Similarly to cancer, specialized membrane domains are involved in pathogen-host interaction. Lipid micro domains mediate pathogen entry, replication, and egress. Thus, lipid depletion can be exploited as a treatment of infections with pathogens like *Plasmodium falciparum*, various bacteria (i.e., *Heliobacter pylori, Mycobacterium tuberculosis)*, and viruses like Influenza A virus, HIV, and SARS-CoV-2. [220,221,222,223]. The druggability of lipid-based pathogen–host interactions has been reviewed extensively elsewhere [224,225,226,227].

## 10. Conclusions

Summed up, there are obvious similarities as well as discrepancies in the regulation of DR signaling in different cell lines derived from different tissues and organisms. On one hand, this can be attributed to evolutionary alterations in the amino acid sequence of the respective proteins, possibly resulting in differential palmitoylation patterns. On the other hand, these effects may be due to the dynamic lipid metabolism and lipid composition of different membranes in these cell lines. Especially cancer cells show high plasticity and can adapt their metabolic state to different environmental conditions. This certainly affects signaling via (death) receptors [228].

When performing experiments in cells or animal models, we have to keep a vigilant eye on how different origins of cells, variable diets of animals, or cell culture supplements (i.e., FCS) affect the experimental outcome. Lipids are connected to various cellular functions, ranging from energy source, membrane structural, or protein recruitment platforms and signaling molecules to substrates for post-translational protein–lipid modification. Thus, different availability of metabolites will affect homeostasis and can result in disease by altering molecular mechanisms, for example by forcing palmitoyl-transferases to use different, more abundant lipids to attach to their substrates.

Differential availability of lipids and their incorporation in the membrane will affect membrane thickness, fluidity, and curvature. Most studies on the association of distinct proteins with lipid rafts, caveolae, or micro-domains have relied on “isolation” of detergent-resistant membranes (DRM) upon treatment of cells with “mild” detergents like 0.1–1% TritonX100 and Brij98 or alternatively rely on alkaline treatment, both followed by ultracentrifugation on sucrose or OptiPrep^TM^ gradients. Subsequently, DRM “marker” proteins (i.e., flotillin-1/-2, caveolin-1/-2, GM1, GM3) are monitored and compared with the protein of interest. Disruption of DRM by using βMCD, nystatin, or filipin is often applied prior to functional analysis. However, these compounds are far from selective. Additionally, “labeling” of lipid rafts using for example cholera toxin B subunit is known to alter and bias membrane composition. Accepting this as state-of-the-art, we have to keep in mind that one cannot discriminate between DRM derived from different subcellular locations or DRM with different lipid compositions. Subcellular pre-fractionation, as we recently suggested, may therefore improve the resolution of such analyses [229]. Various aspects to understand and investigate the function of membrane lipids have recently been reviewed elsewhere [7,230].

Obviously, distinct signaling cascades emanate from different subcellular locations and even distinct micro-domains in membranes. Thus, when interpreting data derived from immune-precipitated (IP) material, it is important to keep in mind that the detergents used in these approaches destroy the lipid composition of membranes. As a result, it is not possible to discriminate IP material whose protein content was soluble before or has been solubilized due to detergents in the lysis buffer.

Usually, 2-Bromopalmitate (2BrP) is used as an inhibitor to analyze palmitoylation of proteins. However, despite having been the “best” inhibitor of palmitoyl transferases, it displays no selectivity for specific PATs. It also inhibits PTE and is a non-selective inhibitor of lipid metabolism. Other inhibitors are cerulenin and tunicamycin; however, these also affect lipid metabolism and glycosylation, respectively. Recently, a novel acrylamide-based PAT inhibitor has been developed, claiming higher potency and less toxicity compared to 2BrP [231]. Similar aspects have to be kept in mind when working with PTE inhibitors. Palmostatin B is frequently used; however, it affects various serine hydrolases. During the last years, inhibitors with higher selectivity and probes to determine enzyme activity have been developed [12,15].

Despite these limitations, palmitoylation, as a reversible lipid post-translational modification, has evolved as an intensively studied research topic over the past two decades. Besides other thoroughly studied PTM like phosphorylation and ubiquitination or proteolysis, palmitoylation regulates signaling via death receptors in a tissue- and cell-type-dependent context. Especially, a better understanding of the spatio-temporal distribution of death receptors within distinct subcellular compartments and membrane environments, as well as the involvement of lipid PTM in the respective regulatory processes, can lead to advances in targeted and personalized therapeutic approaches for a variety of diseases, like cancers, infection, inflammation, cardiovascular diseases, or viral infection. 

## Figures and Tables

**Figure 1 cancers-13-02513-f001:**
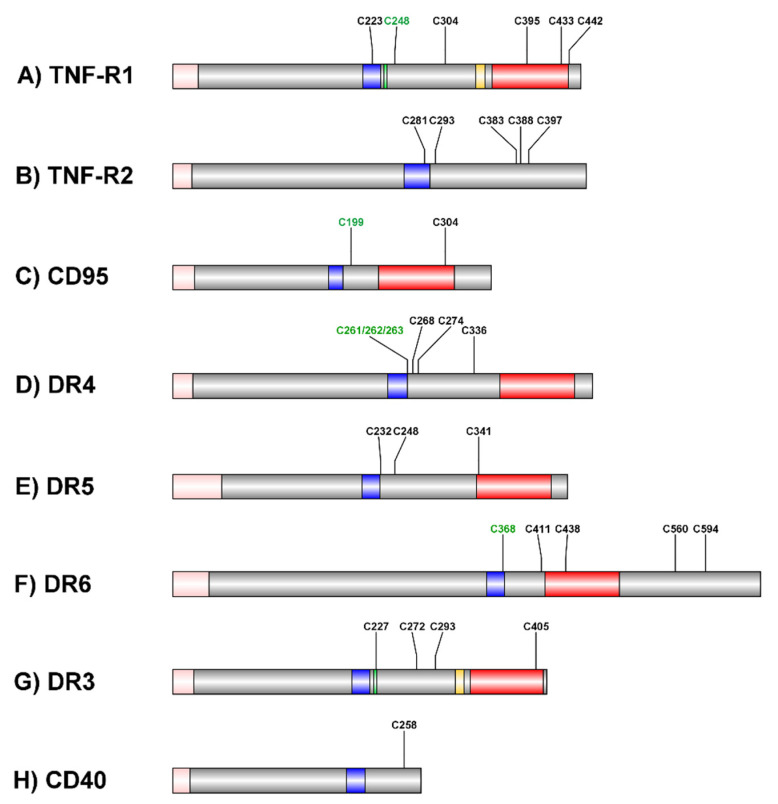
Schematic representation of receptors and their known and putative palmitoylation sites. Signal peptides (SP) are marked in pink, trans-membrane domain (TMD) in blue, and death domain (DD) in red. Intracellular cysteine residues are indicated as known (green) and putative (black) palmitoylation acceptor sites. (**A**) depicts TNF-R1 (UniProt: P19438). C248 is the sole known palmitoylation site. The TNF-R1 internalization domain (TRID) is marked in green; the neutral sphingomyelinase domain (NSD) is marked in orange. (**B**) depicts TNF-R2 (UniProt: P20333). Five possible palmitoylation acceptor cysteine residues are indicated. (**C**) depicts CD95 (UniProt: P25445). C199 is a known human CD95 palmitoylation site. C194 is palmitoylated in murine CD95. (**D**) depicts DR4/TRAIL-R1 (UniProt: O00220). Cysteine residues 261-263 are known palmitoylation acceptor sites. (**E**) depicts DR5/TRAIL-R2 (UniProt: O14763) with three putative palmitoylation sites. (**F**) depicts DR6 (UniProt: O75509). C368 is a known palmitoylation site. (**G**) Depicts DR3 (UniProt: Q93038). Four possible palmitoylation sites are indicated. A putative internalization domain is marked in green; the putative neutral sphingomyelinase domain (NSD) is marked in orange. (**H**) depicts CD40 (UniProt: P25942). The sole intracellular cysteine residue 258 is indicated.

**Figure 2 cancers-13-02513-f002:**
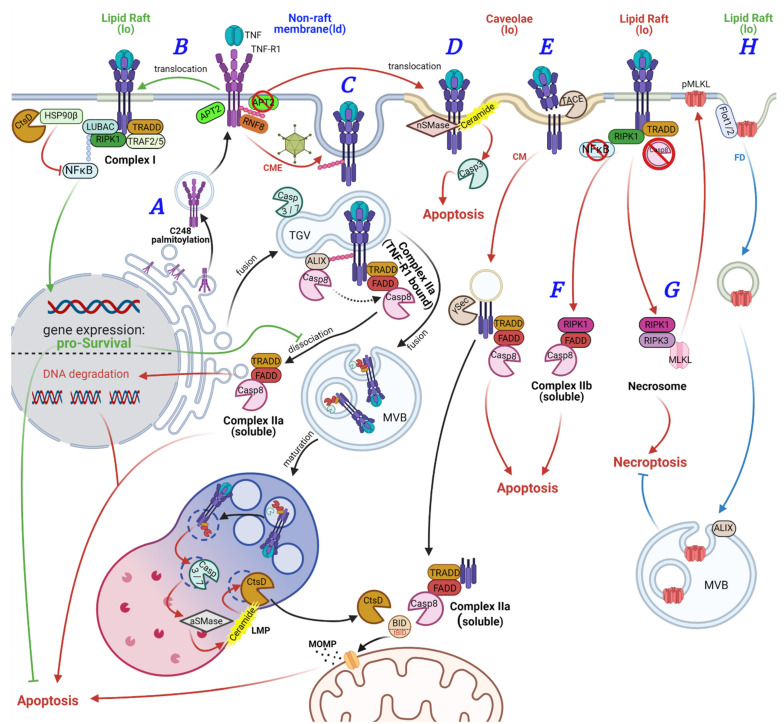
TNF-R1 signaling pathways. (**A**) TNF-R1 palmitoylation at cysteine residue 248 facilitates its transport from the Golgi-apparatus to the non-raft/liquid-disordered (ld) environment of the plasma membrane. (**B**) Binding of TNF to TNF-R1 activates APT2, resulting in partial de-palmitoylation of TNF-R1 and receptor translocation to a liquid-ordered (lo) plasma membrane compartment (lipid raft). In lipid rafts, pro-survival signaling via *complex I* formation and NFκB is initiated. Translocation of NFκB to the nucleus induces transcription of pro-inflammatory/pro-survival genes, inhibiting cell death induction. (**C**) Activated TNF-R1 is ubiquitinated by RNF8, leading to clathrin-mediated endocytosis of the receptor. The resulting TNF-receptosomes fuse with trans-Golgi vesicles (TGV) to recruit, i.e., caspases. At internalized TNF-R1, *complex I* is modified in favor of *complex IIa*. *Complex IIa* can dissociate from the receptor and elicit its apoptosis, inducing function in the cytoplasm. TNF-receptosomes can alternatively be integrated into multivesicular bodies (MVB), which maturate to lysosomal vesicles by acidification. In lysosomes, the receptosome membrane is degraded, and a multi-enzyme cascade results in aSMase activation, ceramide formation, and ultimately Cathepsin D (CtsD) activation and lysosomal membrane permeabilization (LMP). Activated CtsD translocates into the cytoplasm to hydrolyze HSP90β or activates Bid to truncated Bid (tBid). tBid translocates to mitochondria and drives mitochondrial outer membrane permeabilization (MOMP), boosting apoptosis induction by cytochrome C release and apoptosome formation. (**D**) Inhibition of APT2 results in translocation of active TNF-R1 to caveolae, where nSMase is activated, resulting in ceramide formation at the plasma membrane, caspase activation, and apoptosis induction. (**E**) In caveolae, activated TNF-R1 can be shed by TACE followed by clathrin-independent internalization of the TNF-R1 TMD/cytoplasmic tail, where pro-apoptotic complex IIa can be recruited. γ-secretase can release *complex IIa* into the cytoplasm. (**F**) Aberrant NFκB signaling results in formation of RIPK1-dependent pro-apoptotic *complex IIb* and its release into the cytoplasm. (**G**) Inhibition or lack of caspase-8 enzyme activity impedes *complex II* formation in favor of necrosome formation and necroptosis induction. Necroptosis induction involves plasma membrane pore formation upon MLKL phosphorylation, resulting in cell swelling and bursting. (**H**) This can be counteracted by flotillin-dependent endocytosis of pMLKL and sorting via ALIX positive MVB towards lysosomal degradation or release as exosomes. TNF-R1-mediated MAPK or JNK signaling is not shown.

**Figure 3 cancers-13-02513-f003:**
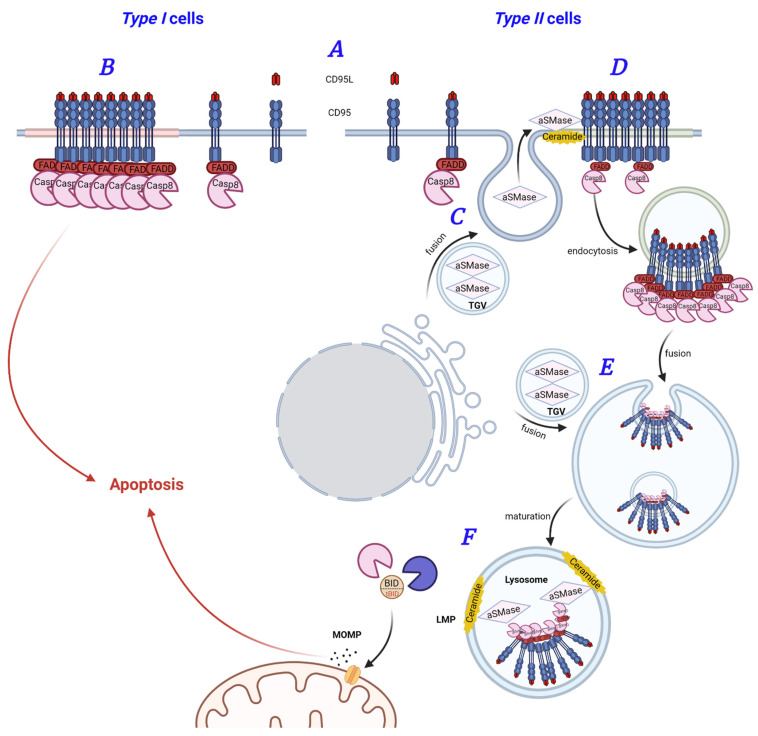
CD95-mediated pro-apoptotic signal transduction. (**A**) Binding of CD95L(FasL) to CD95(Fas) activates the receptor. (**B**) In type I cells, CD95 forms clusters concomitant with rapid and massive DISC formation, directly triggering cell death. (**C**) In type II cells, low amounts of CD95 and active Caspase-8 are required to trigger fusion of trans-Golgi vesicles (TGV) with the PM and secretion of aSMase. (**D**) Lipid raft formation is triggered, propagating the formation of CD95 clusters, followed by receptor internalization. (**E**) CD95-receptosomes fuse with multi-vesicular bodies (MVB) and trans-Golgi vesicles (TGV), allowing acidification of the organelle and acquisition of lysosome characteristics. (**F**) In lysosomes, LMP is induced, resulting in lysosomal enzyme release into the cytosol. Proteases cleave Bid, which promotes mitochondrial outer membrane permeabilization and apoptosis amplification. CD95-mediated NFκB, MAPK, and JNK signaling is not shown.

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
