# Peer review of "Regulation of Death Receptor Signaling by S-Palmitoylation and Detergent-Resistant Membrane Micro Domains—Greasing the Gears of Extrinsic Cell Death Induction, Survival, and Inflammation"

_cancers, 2021, doi:10.3390/cancers13112513_

Round 1
Reviewer 1 Report
This is a very extensive and detailed review, which is at times difficult to follow given the spread of the information. Total text spans 30 pages, plus references. Figures are fine.
An attempt to streamline the text could improve readability and would be recommended.
Author Response
we would like to thank you for your overall very kind comments and appreciation of our review article.
Regarding to comments of you and other reviewers: We decided not to shorten the text as we did not want to omit relevant information. The text is quite elaborate, which is due to the fact that we tried to highlight the most relevant, topic related details of the named proteins in this context.
Reviewer 2 Report
The manuscript by Fritsch J et al “Regulation of death receptor signaling by S-palmitoylation and detergent resistant membrane micro domains - Greasing the gears of extrinsic cell death induction, survival and inflammation” discus the role of proteins palmitoylation in TNF family signal transduction. In this work, the authors reviewed a large amount of experimental data regarding the role of post-translational modification (in particular, palmitoylation) of death receptors (TNFR1, TNFR2, CD95, DR4, DR5 and DR6) and other molecules in signal transduction of the TNF family ligands. In particular, emphasis is placed on the translocation of death receptors into lipid rafts during biochemical modifications and the role of these membrane microdomains in the qualification of apoptosis or cell survival signals. Taken together, the work is very informative and provides a comprehensive understanding of the role of post-translational modification of death receptors based on the data currently published.
Author Response
we would like to thank you for your overall very kind comments and appreciation of our review article.
Reviewer 3 Report
This manuscript entitled “Regulation of death receptor signaling by S-palmitoylation and detergent resistant membrane micro domains - Greasing the gears of extrinsic cell death induction, survival and inflammation” reviews the functional and mechanistic roles for S-palmitoylation as well as the different forms of membrane micro-domains in death receptor-mediated signal. The manuscript is well-written and interesting. However, the authors need to address a few comments.
Comments:
- Symbols are missing. For example, TNF- instead of TNF-alfa (page 1, line 43); NF B instead of NF-kB (page 2, lines 61, 66, and 74, etc…).
- The text from section 1 should be better cited. For example, paragraphs from page 2 lines 77 to 92 don’t have any references and some paragraphs only have references at the end (from page 2/line 93 to page 3/106).
- Since the regulation of death receptor signaling depends on cell type, tissue, and organism when the Authors present a cell line for the first time, they should indicate their cell type (for example, MCF-7 is a breast cancer cell line and HUVEC is an endothelial cell line).
- MOMP is not written out in full in its first use (page 8, line 411).
- The endoplasmic reticulum is abbreviated on page 11 (line 533) but its first use is on page 3 (line 115).
- Section 6 is missing. The numbering of the sections should be revised.
- The captions of figures 1 and 2 are cut and it is not possible to read the whole caption.
Author Response
we would like to thank you for your overall very kind comments and appreciation of our review article.
All changes made are indicated in the text in green or via the ‘tracking’ function.
1. We have to apologize for using ‘symbol’ font. When changing the font during editing, this led to loss of greek symbols.
2. For the first paragraph two, two citations were included.
3. We also now named the origin of all cell lines cited in the text.
4. MOMP abbreviation was corrected accordingly.
5. ER abbreviation was corrected accordingly.
6. The numbering of the sections was corrected.
7. The captions of the figures were corrected. The figures are now provided in the final high resolution, as separate files.
Reviewer 4 Report
This is an extensive and well written review article that summarizes the recent findings on the role of palmitoylation and lipid rafts in TNF, TRAIL and FasL signaling. This review would be of great interest for the readers of Cancers.
Given the difference in palmitoylation patterns and lipid composition of different cell membranes, have lipid rafts been targeted for therapy specifically in:
- Preventing viral or bacterial infections
- Modulating signal transduction in cancer cells to induce cell death.
A paragraph addressing these comments would complement the review.
Author Response
we would like to thank you for your overall very kind comments and appreciation of our review article.
All changes made are indicated in the text in green or via the ‘tracking’ function.
We added another short paragraph, pointing towards the therapeutic potential of the described and related clinical studies. However, we decided to mainly refer to some excellent comprehensive review articles on this topic, as more details would be beyond the scope of this review article.